# Repellency of Carvacrol, Thymol, and Their Acetates against Imported Fire Ants

**DOI:** 10.3390/insects14100790

**Published:** 2023-09-28

**Authors:** Pradeep Paudel, Farhan Mahmood Shah, Dileep Kumar Guddeti, Abbas Ali, Jian Chen, Ikhlas A. Khan, Xing-Cong Li

**Affiliations:** 1National Center for Natural Products Research, The University of Mississippi, University, MS 38677, USA; phr.paudel@gmail.com (P.P.); fshah@olemiss.edu (F.M.S.); gdileepkumar19@gmail.com (D.K.G.); aali@olemiss.edu (A.A.); ikhan@olemiss.edu (I.A.K.); 2Biological Control of Pests Research Unit, USDA-ARS, Stoneville, MS 38776, USA; jian.chen@usda.gov; 3Department of BioMolecular Sciences, School of Pharmacy, The University of Mississippi, University, MS 38677, USA

**Keywords:** *Solenopsis invicta*, *Solenopsis richteri*, *Solenopsis invicta* × *Solenopsis richteri*, minimum repellent effective dose, essential oil

## Abstract

**Simple Summary:**

Imported fire ants are significant pests of urban, agricultural, and medical importance, causing USD billions of annual losses in the United States. Synthetic insecticides are commonly used in their management. The potential adverse effects of synthetic insecticides highlight the need to develop natural-product-based alternatives for fire ant control. Repellants are useful in managing fire ants; for example, repellants can be used to prevent fire ants from invading sensitive areas, such as electrical equipment, nursing homes, and hospitals. In particular, plant-derived natural repellants may provide a safer and more environmentally friendly alternative. This study demonstrates the repellent effects of the plant-essential-oil-derived compounds carvacrol, thymol, and their acetate derivatives against imported fire ants. Carvacrol, a GRAS compound (Generally Recognized As Safe) was the most potent repellent against red, black, and imported fire ants with minimum repellent effective doses of 0.98 µg/g, 7.80 µg/g, and 0.98 µg/g, respectively, followed by thymol, carvacrol acetate, and thymol acetate. Thymol red essential oil containing carvacrol and thymol also showed repellency. These results indicated that carvacrol and thymol as well as essential oils with high contents of carvacrol and/or thymol are potentially useful in managing imported fire ants.

**Abstract:**

In the United States, imported fire ants are commonly referred to as red imported fire ants (*Solenopsis invicta* Buren), black imported fire ants (*S. richteri* Forel), and hybrid imported fire ants (*S. invicta* × *S. richteri*). They are significant pests, and their control heavily relies on synthetic insecticides. The extensive use of insecticides has led to public concern about their potential negative effects on human health and the well-being of wildlife and the environment. As an alternative, plant-derived natural compounds, particularly essential oils (EOs) and their main constituents, show promise as safe and environmentally friendly products for controlling fire ants. Repellants are useful in managing fire ants, and plant-derived natural repellants may serve as a safer and more environmentally friendly option. This study investigates the repellency of EO-derived compounds carvacrol, thymol, and their acetates against imported fire ant workers. The results revealed that carvacrol, a GRAS compound (Generally Recognized As Safe), was the most potent repellent against *S. invicta*, *S. richteri*, and their hybrid, with minimum repellent effective doses (MREDs) of 0.98 µg/g, 7.80 µg/g, and 0.98 µg/g, respectively. Thymol also exhibited strong repellency, with MREDs of 31.25 µg/g, 31.25 µg/g, and 7.8 µg/g, respectively. Furthermore, thyme-red essential oil, characterized by a thymol chemotype containing 48.8% thymol and 5.1% carvacrol, was found to effectively repel the hybrid ants with an MRED of 15.6 µg/g. In contrast, thyme essential oil, characterized by a linalool chemotype lacking thymol and carvacrol, did not exhibit any repellent effect, even at the highest tested dose of 125 µg/g. This study provides the first evidence of the potent repellency of carvacrol and thymol against imported fire ant workers, indicating their potential as promising repellents for fire ant control.

## 1. Introduction

Imported fire ants, including red imported fire ants (*Solenopsis invicta* Buren) (Hymenoptera: Formicidae), black imported fire ants (*Solenopsis richteri* Forel), and hybrid imported fire ants (*S. invicta* × *S. richteri*), are major invasive pest ants in the United States [1]. Fire ants are omnivores and adversely affect humans, wildlife, agriculture, and livestock [2] through their venomous stings and feeding and mound-building habits. Ecological impacts include biodiversity loss by predation and competition with various organisms, including native ant species, which cause changes in ecosystem processes [3,4]. The increasing populations of fire ants impact the growth, harvest, and yield of many crops, soybeans for instance [5], leading to losses measured in USD billions annually [6]. While the red imported fire ant is the dominant species in the southern United States, the hybrid fire ant has a particularly wide distribution in northern Mississippi, Tennessee, Northern Alabama, and northern Georgia [7,8]. Synthetic insecticides are primary tools for fire ant control. However, growing concerns about the negative effects of synthetic insecticides on public health and the environment have increased the need for safe and environmentally friendly products and pest control methods.

Essential oils (EOs), often referred to as “Green Pesticides,” have gained popularity in modern organic agriculture as a safe and environmentally friendly pest control option that is compatible with biological control methods [9]. EOs have been used for decades as insecticides, fumigants, antifeedants, and repellents, with their main constituents being responsible for their efficacy [6,10]. Carvacrol (5-isopropyl-2-methylphenol) and thymol (2-isopropyl-5-methylphenol), the primary constituents of thymol chemotype thyme EOs [11], as well as other EOs from the Lamiaceae, Verbenaceae, Scrophulariaceae, Ranunculaceae, and Apiaceae families, have received significant attention for their pest control properties [12,13,14]. Numerous studies have demonstrated the repellent and insecticidal effects of carvacrol and thymol against various insects [13,15,16,17,18]. Additionally, carvacrol acetate, a naturally occurring compound found in thyme oil [19], has shown superior repellency against unfed *Rhipicephalus sanguineus* sensu lato ticks compared to carvacrol [20]. Thymol acetate, a minor compound naturally found in thyme oil [21], has exhibited a higher toxicity against the beet armyworm than thymol [22]. To the best of our knowledge, there are no reports on the behavioral effects of carvacrol and thymol, as well as their acetates, against imported fire ants. Therefore, this investigation of these four compounds (depicted in Figure 1) against imported fire ants represents a novel and timely research endeavor.

## 2. Materials and Methods

### 2.1. Chemicals, Analysis, and Synthesis

#### 2.1.1. Chemicals

Carvacrol (purity ˃ 98%) was purchased from Tokyo Chemical Industry—TCI (Tokyo, Japan), and thymol (purity ˃ 99%) from Sigma Aldrich (St. Louis, MO, USA). Carvacrol acetate and thymol acetate were synthesized (see below in Section 2.1.3). Two thyme EOs, thyme-red from *Thymus vulgaris* (thymol chemotype, Greek origin) and thyme from *Thymus vulgaris* (linalool chemotype, French origin), were purchased from Edens Garden (Blaine, MN, USA).

#### 2.1.2. GC-MS Analysis

Thyme-red EO and thyme EO were analyzed by GC-MS using an Agilent 7890 B GC system equipped with a 5977A quadrupole mass spectrometer and a 7693 autosampler (Agilent Technologies, Santa Clara, CA, USA). The sample was prepared in GC-MS grade dichloromethane (Sigma-Aldrich) at a concentration of 10 mg/mL. Helium was used as the carrier gas at a flow rate of 1.5 mL/min. In split injection mode, the inlet temperature was set to 280 °C and the split ratio was set to 30:1. The oven temperature program was initially set to 60 °C, held for 2 min, then ramped up to 280 °C at a rate of 6 °C/min, and heated isothermally at 280 °C for 10 min, for a total run time of 77 min. Data acquisition was performed using Agilent MassHunter software (version B.07.06). Compound identification was based on an NIST library search and comparison with reference standards. The main constituents present in thyme-red EO were thymol (48.8%), *p*-cymene (16.3%), γ-terpinene (6.4%), carvacrol (5.1%), caryophyllene (3.1%), linalool (2.4%), eucalyptol (1.4%), α-humulene (1.1%), and (-)-terpinen-4-ol (1.0%), while the major compounds in thyme EO were linalool (71.9%), linalyl acetate (13.4%), and caryophyllene (3.4%). The percentage compositions of other constituents in these EOs are shown in Appendix A and their total ion chromatograms are shown in Appendix A.

#### 2.1.3. Synthesis of Carvacrol Acetate and Thymol Acetate

A mixture of carvacrol (500 mg) and acetic anhydride (2.0 equivalents) in CH_2_Cl_2_ (15 mL) containing 555 µL of triethylamine and a catalytic amount of 4-dimethylaminopyridine (DMAP, 37 mg) was stirred at room temperature for 1 h under N_2_. The reaction mixture was washed with HPLC-grade water (20 mL × 3). The organic phase was dried with anhydrous Na_2_SO_4_ and evaporated to dryness to obtain carvacrol acetate (530 mg). Identification of the product was confirmed by ^1^H and ^13^C NMR spectra (Appendix A). Its purity was determined to be >95% by GC-MS.

Similarly, a mixture of thymol (1000 mg) and acetic anhydride (1.25 mL, 2 eqv.) in CH_2_Cl_2_ (30 mL) containing 1.11 mL of triethylamine and a catalytic amount of DMAP (70 mg) was stirred at room temperature for 2 h under N_2_. The solution was washed with HPLC-grade water (40 mL × 3). The organic phase was dried with anhydrous Na_2_SO_4_ and evaporated to dryness to obtain thymol acetate (1.21 g). Identification of the product was confirmed by ^1^H and ^13^C NMR spectra (Appendix A). Its purity was determined to be >95% by GC-MS.

### 2.2. Ants

Colonies of red imported fire ants, black imported fire ants, and hybrid ants were collected from Tunica County, MS-713; Hernando, DeSotos County, MS 38632 (34°49′56.5″ N 90°12′55.6″ W); Washington County, MS 38748 (33°09′31.2″ N 90°54′56.4″ W); and the University Field Station (University of Mississippi, 15 County Road 2078, Abbeville, MS 38601, USA), respectively. The ant colonies were kept in the laboratory under conditions of 25 ± 2 °C and 50% ± 10% relative humidity and fed with crickets and a 25% honey/water solution. The ants were maintained under laboratory conditions for one month before starting the bioassays. The ant species were identified based on the venom alkaloid and hydrocarbon profiles of the collected individuals [23,24]. The workers of the three ant species were used in this study.

### 2.3. Digging Bioassay

The digging bioassay used in this study for testing repellency has been described in our previous paper [23], which is based on the fact that the fire ant workers always show digging behavior when exposed to an adequate digging substrate including sand. When the sand is treated with a test sample possessing repellent activity, the workers would not dig or dig less, and the repellency of the test sample is measured by comparison of the quantity of sand removed with that of a blank control. Briefly, this bioassay consisted of four 2 mL Nylgene Cryoware Cryogenic vials with caps (Thermo Fisher Scientific, Rochester NY 14825, USA) glued to the bottom of an arena of a 150 mm × 15 mm Petri dish (Fisher Scientific Co., LLC, 2775 Horizon Ridge CT, Suwanee GA 30024, USA) at equal distances. Insect-a-Slip (BioQuip Products 2321 Gladwick Street Rancho Dominguez, CA 90220, USA) coated on the inner side of the arena Petri dish prevented the escape of worker ants. The sand (Premium Play Sand, Plassein International, Longview, TX, USA) was sieved through a 35-mesh USA standard testing sieve (Thomas Scientific, Swedesboro, NJ, USA) to achieve a uniform size of 500 microns, which was then washed with de-ionized water and oven-dried at 150 °C for 6 hr. A fluted aluminum (45 mL size) weighing dish (Fisher Scientific, 300 Industry Drive Pittsburgh, PA 15275, USA) was used to weigh 4.0 g of sand. Each 4.0 g of sand in the aluminum pan was mixed thoroughly with a solution of a test compound in ethanol (400 µL). The solvent was allowed to evaporate at room temperature. Once the solvent evaporated, the sand was moistened by adding 0.6 µL/g of de-ionized water. Treated sand was placed in vials using small spatulas to ensure that no space was left in the vials. The sand used for the control treatment was treated with ethanol only. The vials were then screwed to the caps at the bottom of the Petri dish arena. Fifty workers of imported fire ants were released in the center of the arena Petri dish to access sand under laboratory conditions of 25 ± 2 °C and 50 ± 10% relative humidity. At 24 h post-treatment, the sand from treated vials was collected in aluminum dishes, oven dried at 150 °C for 1 h, and then weighed. A series of dosages were tested, starting from 125 µg/g until the quantity of sand removed became similar to the ethanol control. Each experiment was replicated at least 3 times. DEET (N,N-diethylmeta-toluamide), a well-known insect repellent for its potency and simple synthesis developed by the U.S. Army in 1946 [25], was used as a positive control. The minimum repellent effective dose (MRED) of a test compound is defined as the dose (µg/g) at which the quantity of sand removed is significantly lower than the ethanol control.

### 2.4. Data Analysis

Data were analyzed by using an analysis of variance (ANOVA) (SAS 9.4, 2012) followed by a Ryan–Einot–Gabriel–Welsch multiple range test for mean separation at *p* ≤ 0.05 (SAS 9.4 (2012)).

## 3. Results

As shown in Table 1, carvacrol exhibited potent repellent effects at the doses of 0.98, 7.8, and 0.98 µg/g against red imported fire ants, black imported fire ants, and the hybrid ants, respectively, which are defined as minimum repellent effective doses (MREDs). ThyRmol, the structural analogue of carvacrol, produced MREDs of 31.25, 31.25, and 7.8 µg/g against red imported fire ants, black imported fire ants, and the hybrid ants, respectively. The acetate derivatives of the aforementioned two compounds, carvacrol acetate and thymol acetate, appeared to possess weaker activity, exhibiting MREDs of 31.25, 31.25, and 15.6 µg/g and 62.5, 31.25, and 125 µg/g, respectively, against red imported fire ants, black imported fire ants, and the hybrid ants, respectively. For comparison, DEET gave MREDs of 62.5, 125, and 31.25 µg/g against red imported fire ants, black imported fire ants, and the hybrid, respectively. Therefore, carvacrol represents the most potent compound, followed by thymol, carvacrol acetate, and thymol acetate in terms of the overall repellency against the three tested imported fire ant species.

Two chemotypes of thyme oils, thyme-red EO containing thymol (48.8%) and carvacrol (5.1%) and thyme EO containing linalool (71.9%) and linalool acetate (13.4%), were also tested against the hybrid imported fire ants. The MRED of thyme-red EO was 15.6 µg/g. For thyme EO, in which thymol and carvacrol are not present, no repellent effect was observed at the highest test dose of 125 µg/g.

## 4. Discussion

Given the high impact of imported fire ants on agriculture, human health, and the environment, interest in developing novel repellents is gaining attention. Previously, synthetic insecticides such as bifenthrin and tefluthrin were used to repel fire ants from potting soil [26] and permethrin-impregnated nylon plastics were used to repel fire ants from electrical housings and equipment boards [27]. Due to the adverse effects of synthetic insecticides on human health and their ecological side effects, natural products are of renewed interest [6]. A tremendous effort towards the search for effective and safe alternatives to synthetic insect repellents has led to the identification of phthalates [28], terpenoids [29,30], allylbenzenes [31], aminobenzenes [32], alkylamines [33], and pyrone derivatives [34] with varying degrees of repellency against fire ants.

Our study showed that carvacrol and thymol, two structurally close monoterpene isomers, had remarkable repellent activity against imported fire ant workers. Unlike the superior repellent effect of carvacrol acetate compared to carvacrol in ticks [20], carvacrol acetate demonstrated less repellency in imported fire ant workers. Similarly, thymol acetate showed a much weaker repellency compared to thymol. This indicates that the free hydroxy group on the phenol ring of the two compounds is critical to the fire ant repellency. Although these two acetates are chemically more stable than their parent compounds, their weaker repellency precludes further evaluation of these compounds or their ester analogs against fire ants.

The repellency of carvacrol against imported fire ant workers is approximately 32-, 8-, and 7-fold higher than thymol against red imported fire ants, black imported fire ants, and the hybrid ants, respectively. This seems to be consistent with a previous report describing mosquito repellency of the two compounds against *Aedes albopictus* [15], where carvacrol is about three-fold more potent than thymol. In addition, an investigation of repellency against the poultry red mite (*Dermanyssus gallinae*) showed that carvacrol was superior to thymol [35]. Carvacrol is larvicidal against *A. albopictus* [36], insecticidal against the brown-winged cicada (*Pochazia shantungensis*) [16], and fumigant against rusty grain beetles (*Cryptolestes ferrugineus*) [37]. Taken together, our data further support that carvacrol is a promising pest control agent [38].

Thymol was initially registered as a pesticide in the United States in 1964 for use as a repellent (EPA-738-F-93-010). It repels vertebrate pests by a non-toxic mode of action, which should not result in unreasonable adverse effects on human health or the environment. In recent years, thymol’s repellent and insecticidal activities against mosquitoes have been studied extensively [35,39,40]. Its insecticidal and genotoxic effect on the fruit fly (*Drosophila melanogaster*) [41] has also been noted. The potent repellency of thymol identified in this study makes it a strong repellent candidate for fire ant control.

Our study has also shown that thymol-red EO containing carvacrol and thymol exhibited potent repellency against hybrid imported fire ants, and these two compounds are likely the active compounds responsible for the observed repellency. The thymol-chemotype of thyme EOs or many other EOs with high contents of carvacrol and/or thymol [10] may exert a potent repellent effect against imported fire ants.

It is worth noting that carvacrol and thymol are listed as food additives by the US Food and Drug Administration, and they are “Generally Recognized As Safe” (GRAS) in food when used in the minimum quantity to produce their intended effect. In 2021, the Expert Panel of the Flavor and Extract Manufacturers Association further affirmed the GRAS status of carvacrol and thymol and several EOs containing these compounds under their conditions of intended use as flavor ingredients [42]. This information is of particular significance when carvacrol- and thymol-based new agricultural products are developed in terms of safety and environmental concerns.

## 5. Conclusions

This study has demonstrated the repellency of carvacrol and thymol against imported fire ant workers. This is the first report of the repellency of the two compounds against imported fire ants. In particular, carvacrol is a more promising candidate due to its remarkable repellent effects towards all the three fire ant species tested. In the future, more structural analogues of carvacrol and thymol may be synthesized for structure–activity relationship studies. In addition, formulation studies, e.g., microencapsulation for long-lasting repellency, may be conducted to develop carvacrol- and thymol-based products for fire ant control.

## Figures and Tables

**Figure 1 insects-14-00790-f001:**
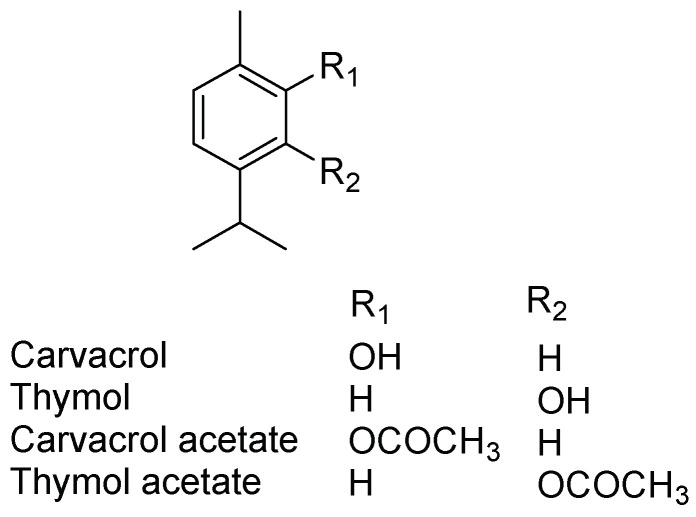
Chemical structures of tested compounds.

**Table 1 insects-14-00790-t001:** Repellency of carvacrol, thymol, and their acetates against the workers of imported fire ant in a digging bioassay.

Dose (µg/g)	Mean ± SE *	*F*-Value	*p*-Value	Mean ± SE *	*F*-Value	*p*-Value	Mean ± SE *	*F*-Value	*p*-Value
	Red imported fire ants (*Solenopsis invicta*)			Black imported fire ants (*Solenopsis richteri*)			Hybrid imported fire ants (*S. invicta* × *S. richteri*)		
Carvacrol									
Control	1.10 ± 0.05 ^a^	420.32	<0.0001	1.18 ± 0.22 ^a^	27.57	<0.001	2.10 ± 0.10 ^a^	34.55	<0.0001
125	0.00 ± 0.00 ^b^			0.00 ± 0.00 ^b^			0.03 ± 0.02 ^b^		
62.5	0.00 ± 0.00 ^b^			0.00 ± 0.00 ^b^			0.26 ± 0.24 ^b^		
31.25	0.01 ± 0.01 ^b^			0.00 ± 0.00 ^b^			0.62 ± 0.18 ^b^		
Control	1.25 ± 0.09 ^a^	16.8	0.0008	2.88 ± 0.11 ^a^	58.99	<0.0001	2.14 ± 0.16 ^a^	7.43	0.0106
15.6	0.13 ± 0.06 ^b^			0.03 ± 0.18 ^c^			0.70 ± 0.35 ^b^		
7.8	0.25 ± 0.15 ^b^			1.39 ± 0.15 ^b^			0.59 ± 0.22 ^b^		
3.9	0.52 ± 0.16 ^b^			2.05 ± 0.13 ^a^			1.12 ± 0.26 ^b^		
Control	0.74 ± 0.03 ^a^	6.64	0.0146	2.65 ± 0.48 ^a^	1.52	0.2821	1.76 ± 0.11 ^a^	5.55	0.0235
1.95	0.20 ± 0.14 ^bc^			1.99 ± 0.47 ^a^			1.03 ± 0.17 ^b^		
0.98	0.05 ± 0.30 ^c^			2.36 ± 0.16 ^a^			0.98 ± 0.16 ^b^		
0.49	0.62 ± 0.21 ^ab^			2.65 ± 0.07 ^a^			1.49 ± 0.19 ^ab^		
Thymol									
Control	1.00 ± 0.09 ^a^	22.51	<0.001	1.20 ± 0.34 ^a^	11.82	0.003	2.51 ± 0.37 ^a^	16.92	0.0008
125	0.10 ± 0.10 ^b^			0.00 ± 0.00 ^b^			0.04 ± 0.04 ^b^		
62.5	0.13 ± 0.13 ^b^			0.00 ± 0.00 ^b^			0.68 ± 0.35 ^b^		
31.25	0.12 ± 0.02 ^b^			0.00 ± 0.00 ^b^			0.39 ± 0.18 ^b^		
Control	0.99 ± 0.26 ^a^	3.52	0.068	0.89 ± 0.31 ^a^	1.03	0.43	2.26 ± 0.06 ^a^	5.46	0.0245
15.6	0.30 ± 0.14 ^a^			0.28 ± 0.18 ^a^			1.39 ± 0.13 ^b^		
7.8	0.49 ± 0.08 ^a^			0.52 ± 0.28 ^a^			1.10 ± 0.36 ^b^		
3.9	0.53 ± 0.04 ^a^			0.44 ± 0.22 ^a^			1.49 ± 0.17 ^ab^		
Carvacrol acetate									
Control	1.25 ± 0.06 ^a^	14.4	<0.001	1.21 ± 0.17 ^a^	44.9	<0.001	1.87 ± 0.25 ^a^	16.09	0.0009
125	0.03 ± 0.03 ^b^			0.00 ± 0.00 ^b^			0.10 ± 0.10 ^b^		
62.5	0.19 ± 0.14 ^b^			0.00 ± 0.00 ^b^			0.33 ± 0.17 ^b^		
31.25	0.48 ± 0.23 ^b^			0.04 ± 0.04 ^b^			0.47 ± 0.27 ^b^		
Control	0.90 ± 0.05 ^a^	1.86	0.21	0.77 ± 0.38 ^a^	0.904	0.48	2.18 ± 0.07 ^a^	4.6	0.0374
15.6	0.37 ± 0.11 ^a^			0.25 ± 0.25 ^a^			1.29 ± 0.19 ^b^		
7.8	0.48 ± 0.24 ^a^			0.19 ± 0.14 ^a^			1.64 ± 0.20 ^ab^		
3.9	0.60 ± 0.19 ^a^			0.62 ± 0.35 ^a^			1.74 ± 0.19 ^ab^		
Thymol acetate									
Control	0.91 ± 0.26 ^a^	4.79	<0.001	1.30 ± 0.94 ^a^	126.80	<0.001	1.59 ± 0.3 ^a^	4.09	0.0492
125	0.11± 0.11 ^b^			0.01 ± 0.01 ^b^			0.30 ± 0.06 ^b^		
62.5	0.18 ± 0.11 ^b^			0.06 ± 0.06 ^b^			0.67 ± 0.42 ^ab^		
31.25	0.33 ± 0.13 ^ab^			0.00 ± 0.00 ^b^			1.13 ± 0.19 ^ab^		
Control	-			1.42 ± 0.18 ^a^	4.37	0.042	-	-	-
15.6	-			0.63 ± 0.31 ^ab^			-	-	-
7.8	-			0.25 ± 0.18 ^b^			-	-	-
3.9	-			0.32 ± 0.31 ^b^			-	-	-
DEET									
Control	1.43 ± 0.19 ^a^	16.24	0.001	1.38 ± 0.25 ^a^	8.9	0.006	1.58 ± 0.11 ^a^	9.71	0.0050
125	0.08 ± 0.04 ^c^			0.00 ± 0.00 ^b^			0.42 ± 0.25 ^b^		
62.5	0.74 ± 0.18 ^b^			1.22 ± 0.04 ^a^			0.87 ± 0.13 ^b^		
31.25	1.14 ± 0.10 ^ab^			0.79 ± 0.33 ^a,b^			0.84 ± 0.04 ^b^		
Control	-			-			1.26 ± 0.19 ^a^	0.24	0.8700
15.6	-			-			0.98 ± 0.49 ^a^		
7.8	-			-			1.37 ± 0.28 ^a^		
3.9	-			-			1.16 ± 0.29 ^a^		

* Sand removed is in grams. Means within a column in an experiment followed by the different letter are significantly different (Ryan–Einot–Gabriel–Welsch multiple range test; *p* ≤ 0.05). The data analysis is based on a comparison of different doses with their respective controls within each experiment. The treatment groups within each experiment consisted of three doses and a control.

## Data Availability

The data presented in this study are available on request from the corresponding author.

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
