# Peer review of "Repellency of Carvacrol, Thymol, and Their Acetates against Imported Fire Ants"

_insects, 2023, doi:10.3390/insects14100790_

Round 1

Reviewer 1 Report

The manuscript entitled "Repellency of carvacrol, thymol, and their acetates against imported fire ants”, submitted for publication to insects, it is interesting, giving useful information useful in managing the imported fire ants. It is considered important to tested essential oils and other natural product-based alternatives materials for controlling the insect population, to reduce the use of the synthetic insecticides and to avoid their disadvantages.

The main question addressed is the possibility of use of safe and environment friendly essential oils to control the imported fire ants. This will contribute to reduction of the extensive use of insecticides, that have negative effects on human health and environment. It is a topic having an increasing interest, as it is considered an alternative method of chemical control, promising effectiveness to insect control. This manuscript doesn’t totally complete a gap in this topic, but it enriches the existing information. The chemical compounds used in this study, are tested to the imported fire ants, that cause damages to the growth, harvest, and yield of many crops in USA. The materials and methods of the study, are considered appropriate for the experiments that made and ensured safe results.  The conclusions consistent with the evidence and arguments presented, answering to the main question posed. The references are appropriate and focused to the target of the study. There are not additional comments on tables and figures, as they impress clearly the data of the experiments.

Author Response

We appreciate the reviewer's efforts and positive comments on our manuscript!

Reviewer 2 Report

It was a pleasure reading the manuscript titled 'Repellency of carvacrol, thymol, and their acetates against imported fire ants' by Paudel et al. They describe of prospecting certain repellents to the important fire ants that is a major problem in the US and other countries where it is invasive.

Although the manuscript is well written and presented for a Communication, it is too basic for the reasons I have provided below. I suggest that more experiments and analysis will add to the impact of the paper and the journal as a whole.

1. The authors have tested the repellency only with 'Digging Bioassay'. Although a valid method, it should be accompanied by other repellency tests to prove "true" repellency. Foraging repellency may be the best method as ants that are exposed to repellency would mainly be the 'foragers' while the other ants are in the nests and may not be exposed to repellents.

2. The authors use DEET as a positive control. Does DEET really repel agitated fire ants. Handling of ants by the authors during experiments would have agitated the ants to a certain degree and DEET is not a good repellent against agitated ants? Rather, DEET is a good repellents against foraging ants. This was one of the reason I asked for a foraging repellency test.

3. In such studies it is always better to go with increased concentration/dose ranges and then subjected to the means to probit analysis to analyze the ED/EC (Effective doses/Effective concentration) before describing the MRED. Without an ED value, how valid can an MRED be? The range of the concentrations of the compounds is too narrow (just 3 in this manuscript) for reporting ED or MRED? This may not present a valid MRED read.

4. One other point I would like to make is that repellents mostly repel foraging fire ants. However, spraying repellents will not deter agitated fire ants from biting or stinging. How have the authors accounted for this behavior of fire ants in this study?

I would suggest the authors to plan more detailed experiments to overcome the above concerns. I understand this is a communication and not a full length article, however, this manuscript can be submitted as a full length article with these concerns addressed. Keeping this in mind I would suggest a "Major revision".

All the best to the authors.

Reviewer 3 Report

This study investigates the repellency of 4 essential oil-derived compounds, carvacrol, thymol, and their acetates against red/black/hybrid imported fire ant workers. The aim of this study was straightforward, the experimental design was rational. The results provide evidence of potential use as promising repellents in control of imported fire ants. 

Author Response

(The authors gave the same response as above.)

Reviewer 4 Report

The paper is of high interest. The manuscript Insects-2566355 “Repellency of carvacrol, thymol, and their acetates against imported fire ants” investigated that the repellency of EO-derived compounds, carvacrol, thymol, and their acetate against imported fire ant workers. In this manuscript, the authors explored the potential of carvacrol and thymol as promising repellents for fire ant control. However, there are still some comments that needed to be addressed.  

1. Line 104-109: As the GC-MS results as shown in the manuscript, carvacrol and thymol are not the two most abundant compositions. Why did you choose these two compounds to study on imported fire ants?  

2. Line 137: How did you choose concentrations range of carvacrol and thymol?

3. Line 153: “Each 4.0 g of sand in the aluminum pan was mixed thoroughly with a solution of a test compound……”, what is the amount of solution? Whether it can be mixed well with sand

4. Line 154: “……with a solution of a test compound in ethanol 154 (400 µL)”, please indicate the concentration of ethanol in the solution.

5. Line 195: In the sentence “……, no repellent effect was observed at the highest test dose of 125 µg/g” and Table 1, there seems to be no pattern between dose and repellent effect, so how to accurately confirm which concentration is the most effective?

6. In Table 1, the difference analysis in data analysis, whether it is a comparison of different concentrations within different treatment groups, or a comparison of all data together or something else, needs to be explained clearly at the bottom of the Table.

7. In Table 1, the letters “a, b, c, d” between the means in black imported ants' group at the concentrations “Control, 15.6, 7.8, 3.9”, should be wrong, please confirm it.

8. Line 235: Whether economic benefits have been calculated?

The article provides elaborate and interesting data. I do think that the manuscript contains important issues, interesting approaches, and techniques, which can provide a new strategy about imported fire ant workers by using repellency of green compounds, carvacrol and thymol. I consider this manuscript suitable for publication after suggested corrections in the Insects.

There has minor editing of English language required.

Reviewer 5 Report

This paper evaluated the repellent effects of the plant essential oil-derived compounds, carvacrol, thymol, and their acetate derivatives, against imported fire ants. This study provides the first evidence of the potent repellency of carvacrol and thymol against imported fire ant workers, indicating their potential for fire ant control. The overall structure of the article is relatively complete and the amount of data is sufficient. However, there are still some minor problems that need to be revised before publication.

1.     L23-25: The results of the repellent effects can be briefly displayed in the Simple Summary to support this conclusion

2.     Keywords: Suggest add keywords such as repellants, carvacrol, and thymol

3.     L46: “Solenopsis invicta × Solenopsis richteri;;” change to “Solenopsis invicta × Solenopsis richteri;”

4.     L53: There is an extra space after the word “building”, please delete

5.     L84: “Figure 1. .” change to “Figure 1.”

6.     Section 2.6. Suggest make a simple model diagram to show how to carry out the digging bioassay used in this study for testing repellency of fire ants

7.     Table 1: What is the basis for setting different concentrations gradients of carvacrol, thymol, and their acetates in the repellency bioassay of imported fire ant

8.     Table 1: What does the superscript “a,b” mean?  Should it be changed to “ab”?

9.     L242: There is an extra space after the word “and”, please delete

10.  L251: There is an extra space after the word “demonstrated”, please delete

Round 2

Reviewer 2 Report

I have read the authors rebuttal they have provided, but I am not fully convinced. As this manuscript is a "Communication" format it may be published.

However, I would recommend the authors to include other bioassays like olfactometer, contact chemo-reception assays in their future work to understand more on the repellent's properties. It would be interesting to understand if the repellency is by olfaction or contact-mediated chemoreception. However, I commend the authors for this interesting work.

Author Response

We thank this reviewer's insightful comments and will consider performing this kind of assays in the future. 

Reviewer 3 Report

comments for the manuscript revision 1

This study investigates the repellency of 4 essential oil-derived compounds, carvacrol, thymol, and their acetates against red/black/hybrid imported fire ant workers. The aim of this study was straightforward, the experimental design was rational. The results provide evidence of potential use of these chemicals as promising repellents in control of imported fire ants. Before this manuscript is accepted, probably as type of brief report, authors are encouraged to improve the manuscript by addressing following concerns: 

1) Ant identification (Page 3, Lines 13--135): besides venom alkaloid and hydrocarbon profiles, are there other identifying ways for distinguish 3 species (hybrid), such as morphological, genetic (COI by using PCR)characteristics?

2)In Materials and Methods, Page 3, Line 86-125, section 2.1 to section 2.4 can be merged into 2.1 Chemicals, analysis, and synthesis, including 2.1.1 chemicals, 2.1.2 analysis, 2.1.3, acetate synthesis.

3) In Results, Table 1 is not the suitable way in showing differences among repellency effects of chemicals for 3 ant species (hybrid). Figures may be more suitable?

3) 5 Conclusions: this part can be extended a little to provide specific conclusions based on the results and discussion, such as which chemicals are promising for future use to which ant species (if available), in which doses, etc.

Round 3

Reviewer 3 Report

Revisied version generally followed the reviewer's comments for improvement or some explanations were given. However,  the authors are not willing to revise the conclusion part. I would like to keep my comments that conlusion should provide some information about your results, rather than just discussion or prospects.
